

# Transmission potential of mpox in mainland China, June-July 2023: estimating reproduction number during the initial phase of the epidemic

Andrei R. Akhmetzhanov[1,2] and Pei-Hsuan Wu[1]

[1] Institute of Epidemiology and Preventive Medicine, College of Public Health, National Taiwan University, Taipei, Taiwan
[2] Global Health Program, College of Public Health, National Taiwan University, Taipei, Taiwan

## ABSTRACT

Despite reporting very few mpox cases in early 2023, mainland China observed a surge of over 500 cases during the summer. Amid ambiguous prevention strategies and stigma surrounding mpox transmission, the epidemic silently escalated. This study aims to quantify the scale of the mpox epidemic and assess the transmission dynamics of the virus by estimating the effective reproduction number ($R_e$) during its early phase. Publicly available data were aggregated to obtain daily mpox case counts in mainland China, and the $R_e$ value was estimated using an exponential growth model. The mean $R_e$ value was found to be 1.57 (95% credible interval [1.38–1.78]), suggesting a case doubling time of approximately 2 weeks. This estimate was compared with $R_e$ values from 16 other countries' national outbreaks in 2022 that had cumulative case count exceeding 700 symptomatic cases by the end of that year. The $R_e$ estimates for these outbreaks ranged from 1.13 for Portugal to 2.31 for Colombia. The pooled mean $R_e$ was 1.49 (95% credible interval [1.32–1.67]), which aligns closely with the $R_e$ for mainland China. These findings underscore the need for immediate and effective control measures including targeted vaccination campaigns to mitigate the further spread and impact of the epidemic.

## INTRODUCTION

On 9 August 2023, the China Centers for Disease Control and Prevention (CDC) announced 491 cases in mainland China in July, representing a marked escalation from 106 cases in June and a five-fold increase (*China Centers for Disease Control, 2023b*). Although concerning, this surge was largely anticipated by health experts (*Murphy, 2023*; *Yang, 2023*). Amid ambiguous prevention strategies and the stigmatization of specific patient groups—especially men who have sex with men (MSM)—the mpox epidemic intensified unnoticed.

Mpox is a viral infection mainly transmitted through direct skin-to-skin contact (*The United States Centers for Disease Control and Prevention, 2023*). The global mpox outbreak of 2022–2023 was primarily driven by sexual transmission within MSM networks

Corresponding author
Andrei R. Akhmetzhanov,
akhmetzhanov@ntu.edu.tw

(*Thornhill et al., 2022*). With limited spillover to other populations, the 2022 outbreaks predominantly affected high-risk MSM communities (*Endo et al., 2022*; *Murayama et al., 2023*), making these communities primary targets for subsequent vaccination campaigns (*Chitwood et al., 2023*; *Vairo et al., 2023*).

The global transmission of mpox was evident in 2022, but it was not until 2023 that Western Pacific nations, including Japan, South Korea, and Taiwan, saw significant case counts (*Chan & Lee, 2023*; *Endo, Jung & Miura, 2023*; *Kim et al., 2023*; *Nishiura et al., 2023*). By the end of 2022, many of these countries had minimal or no local mpox transmissions, possibly due to ongoing coronavirus disease 2019 (COVID-19)-related travel and social restrictions (*Akhmetzhanov, Ponce & Thompson, 2022*). Given the considerable MSM population in mainland China, the uncontrollable spread of mpox presents a significant public health concern. An effective mpox vaccine exists, however, challenges such as limited accessibility and disease-related stigma have hampered its broad dissemination (*Murphy, 2023*), underscoring the need to understand mpox transmission dynamics in mainland China and predict the evolution of future outbreaks.

Epidemics typically follow a consistent transmission cycle: introduction, escalation, peak, and eventual decline (*Linton et al., 2022*). The initial stages, often marked by exponential growth, offer important insights into understanding of disease spread dynamics. It is crucial to estimate the effective reproduction number, $R_e$, which indicates the average number of secondary transmissions per primary case, during the period in which interventions, behavioral shifts, or acquired immunity have not yet made an impact.

Prior $R_e$ evaluations for the 2022 national mpox outbreaks have revealed variability (*Brand et al., 2023*; *Du et al., 2022*; *Liu et al., 2023*; *Yang et al., 2023*). Using a sub-exponential growth model, *Du et al. (2022)* estimated that the $R_e$ ranged from 1.02 in Portugal to 1.95 in the United States. Conversely, other reports have noted higher $R_e$ values. For example, for the United Kingdom, $R_e$ values of 2.32 and 3.14 were reported (*Liu et al., 2023*; *Yang et al., 2023*), compared with a value of 1.18 estimated by *Du et al. (2022)*. Whereas these studies (*Liu et al., 2023*; *Yang et al., 2023*) applied variations of a simple susceptible–exposed–infected–recovered model, they also relied on case counts that were classified by the date of reporting. Such an approach can result in an upward bias in estimates during the exponential growth phase because it may reflect not only actual transmission dynamics but also the recognition of an outbreak and the discovery of infection clusters. Administrative batching in case reporting can further bias the estimation of $R_e$. Nevertheless, a recent UK mpox study (*Brand et al., 2023*) employed a detailed, more complex susceptible–exposed–infected–recovered model that factored in potential biases, generating $R_e$ values that ranged from 3.02 to 5.16 during the initial stage of the outbreak. Given diverse factors, including cultural differences, differences in outbreak response, and challenges with data collection, identifying how these estimates align with $R_e$ in mainland China is crucial.

In this study, we analyzed data from the World Health Organization (WHO) and other governmental sources to enhance our understanding of mpox transmission in mainland China (*Taiwan Centers for Disease Control, 2023*; *The Government of the Hong Kong Special Administrative Region, 2023*; *World Health Organization, 2023*). By focusing on

case counts solely attributed to mainland China and estimating the $R_e$ during the initial phase of the outbreak, we aimed to compare counts with estimates from 2022 and project potential case counts for the near future. Our ultimate goal was to inform ongoing public health interventions, and potentially optimize efficiency of these interventions, while also contributing to global epidemic preparedness. A preliminary version of this text was previously published as a preprint (*Akhmetzhanov & Wu, 2023*).

## METHODS

### Data collection

This study used three primary data sources. First, daily counts of confirmed mpox cases in mainland China, Taiwan, and the Hong Kong Special Administrative Region were extracted from the WHO global mpox dashboard (2022–2023) as of 15 August 2023 (*World Health Organization, 2023*). These counts were categorized by the date of symptom onset, followed by the date of diagnosis and the date of reporting. Second, a dataset of the daily counts that was confirmed in Taiwan, categorized either by the date of symptom onset or the date of reporting, was retrieved from the National Infectious Disease Statistics System of the Taiwan Centers for Disease Control (CDC) on 23 August 2023 (*Taiwan Centers for Disease Control, 2023*). Third, a line list of cases that were confirmed in Hong Kong was compiled using press releases from the Center for Health Protection of the Department of Health of the Government of Hong Kong Special Administrative Region on 23 August 2023 (*The Government of the Hong Kong Special Administrative Region, 2023*). The WHO data recorded 859 total cases in mainland China, Taiwan, and Hong Kong. These cases were classified as follows: 615 by the date of symptom onset, 225 by the date of reporting, and 19 by the date of diagnosis. The datasets from Taiwan and Hong Kong recorded 292 and 33 cases, respectively; these cases were all classified by the date of symptom onset. Notably, one patient in Hong Kong lacked a definitive date for symptom onset. The report indicated that this patient began showing symptoms at the end of June. The patient was reported in August 2023 and was unlikely to have been accounted in the WHO dataset; therefore, the patient was omitted from our analysis.

To extract cases that were confirmed solely in mainland China, the combined daily counts from the Taiwan and Hong Kong datasets were subtracted from the daily counts recorded in the WHO dataset. In the process, two assumptions regarding data decomposition in the WHO dataset were investigated.

First, we assumed that all counts from Taiwan were reported by the date of symptom onset in the WHO dataset. However, upon subtracting these counts from the WHO data, the resulting daily counts yielded predominantly negative values before 1 June 2023. Subsequently, only one date (21 June 2023) appeared with a negative case count, when the WHO dataset recorded a daily count of two cases, whereas Taiwan and Hong Kong reported three and no cases, respectively. Therefore, we deduced that our initial assumption regarding the integration of Taiwan case count data into the WHO dataset was likely different.

Thus, we hypothesized that all case counts from Taiwan were incorporated into the WHO dataset by their respective dates of reporting to the WHO. Upon examining the

WHO reporting pattern, it became evident that most cases that were defined by their reporting dates—specifically May 2, 9, 16, 23, 30; June 6, 20, 27; and July 4—were recorded at equidistant intervals of 7 days. A single deviation from this pattern was noted: the inclusion of June 8 with the exclusion of June 13 (Fig. S1). Assuming that the reporting date in Taiwan had a 1-day lag to its confirmation date, the case counts in Taiwan aligned almost perfectly with the counts of cases by reporting date in the WHO dataset (Fig. S2). Although case counts from the Taiwan dataset predominantly fell below the WHO dataset counts, matches were observed for the last five dates: June 8, 20, and 27, and July 4 and 11. This observation led us to the conclusion that the latter approach of imputing data from Taiwan to the WHO dataset was plausible.

Moreover, the upper bound of the exponential growth phase was set to $t_{up} = 5$ July 2023, indicating the peak of observed daily counts of symptomatic cases at 21. The exponential growth phase was investigated by considering a 45-day period and placing a lower bound at $t_{lo} = 21$ May 2023. The robustness of the estimates was explored in the sensitivity analysis by varying the dates for both boundaries, which ranged from 16 to 26 May 2023 for the lower bound and 30 June to 10 July 2023 for the upper bound. The duration of the exponential growth phase that was considered in the sensitivity analysis was thus varied between 35 and 55 days.

## Statistical framework

### Reporting delay distribution

Given that all cases with unidentified symptom onset dates were back-projected to their estimated date of symptom onset in the subsequent analysis, estimating the reporting delay distribution was crucial. This distribution measured the period from the date of symptom onset to either the date of reporting or the date of diagnosis. For this estimation, we utilized the Hong Kong line list data, which is composed of 33 records, and a cutoff time of $T = 23$ August 2023.

Each of the reporting delays (time to diagnosis or time to reporting) was fit to the generalized gamma distribution (GGD), which is defined by three parameters: shape ($Q$), location ($M$), and scale ($S$). The choice of GGD was guided by its flexibility and its capability to represent three commonly used distributions: gamma (when $Q = S$), Weibull (when $Q = 1$), and log-normal (when $Q = 0$) (*Hoffmann & Alsing, 2023*). The mathematical inference for the GGD was based on the following: if the random variable $x$ follows a GGD, then the transformation $Q \exp(Q^{-2} z)$, where $z = (\log(x) - M)/S$, adheres to a gamma distribution with shape $Q^{-2}$ and scale 1.

In particular, $O_i$ represented the date of symptom onset, and $D_i$ (where $O_i \leq D_i$) was the date of reporting or diagnosis, extracted from the Hong Kong line list data, with $i = 1, \ldots, 33$. We assumed that the respective times of symptom onset ($o_i$) and reporting or diagnosis ($d_i$) were uniformly distributed within their intervals as follows:

$$o_i \sim \text{Uniform}(O_i, O_i + 1 \text{ day}),$$
$$d_i \sim \text{Uniform}(\max(\{o_i, D_i\}), D_i + 1 \text{ day}), \tag{1}$$

where the symbol "$\sim$" implies "is distributed as." The difference $d_i - o_i$, representing the observed delay, conforms to the following truncated gamma distribution:

$$y_i \mid y_i \leq Y_i \sim \text{Gamma}(\text{shape} = Q^{-2}, \text{scale} = 1), \quad (2)$$

where: $y_i = Q \exp(Q^{-2} z_i)$, $Y_i = Q \exp(Q^{-2} Z_i)$, and $z_i = (\log(d_i - o_i) - M)/S$, $Z_i = (\log(T - o_i) - M)/S$.

The likelihood is represented by the form:

$$L_{\text{rep}}(\theta \mid \{d_i, o_i, T\}) = \prod_i \frac{\gamma(y_i \mid \text{shape} = Q^{-2}, \text{scale} = 1)}{\Gamma(Y_i \mid \text{shape} = Q^{-2}, \text{scale} = 1)}, \quad (3)$$

where $\gamma$ and $\Gamma$ are the probability density function and cumulative density function of the gamma distribution, respectively. The parameters $\theta$ are assumed to have weakly-informative priors: $\log Q, M, \log S \sim \text{Normal}(\text{mean} = 0, \text{SD} = 1)$.

### Reconstructed case counts

Given that not all of the cases were categorized by their date of symptom onset, it was essential to back-project the cases identified solely by their date of diagnosis or reporting to ascertain their dates of symptom onset.

Therefore, $n_t$ was the number of cases that were symptomatic on day $t$, counting from day 1 on $t_{lo}$. Similarly, $m_t^r$ and $m_t^d$ represented the cases with unknown symptom onset dates but with identified reporting and diagnosis dates, respectively. Any non-zero count $m_t^*$ (where $* := \{r, d\}$) was then back-projected using sampling from a multinomial distribution as follows:

$$\left\{ m_{t-s,t}^*; 0 \leq s \leq t \right\} \sim \text{Multinomial}(\text{size} = m_t^*, \text{probs} = \{f_t^*(s \mid \theta^*)\}). \quad (4)$$

In the above, $f_t^*(x \mid \theta^*)$ describes the discretized reporting delay distribution as follows:

$$\begin{aligned} f_t^*(0 \mid \theta^*) &= \Gamma(0.5 \mid \theta^*), \\ f_t^*(s \mid \theta^*) &= \Gamma(s + 0.5 \mid \theta^*) - \Gamma(s - 0.5 \mid \theta^*), \quad s = 1, \ldots, t-1, \\ f_t^*(t \mid \theta^*) &= 1 - \sum_{0 \leq s \leq t-1} f_t^*(s \mid \theta^*) = 1 - \Gamma(t + 0.5 \mid \theta^*). \end{aligned} \quad (5)$$

The final equation in Eq. (5) indicates the likelihood of a case being reported on day $t$ but manifesting symptoms before day 1 (*i.e.*, before $t_{lo}$). To compute the overall daily case counts, the following summation was employed:

$$c_t = n_t + \sum_{x \geq t} m_{t,x}^r + \sum_{x \geq t} m_{t,x}^d, \quad 1 \leq t \leq T. \quad (6)$$

### Effective reproduction number ($R_e$) during the initial phase of the epidemic

Earlier, we defined the exponential growth phase using the date range $t_{lo} \leq t \leq t_{up}$. The $R_e$ for the initial phase of the epidemic was estimated by analyzing the case counts ($c_t$) with the presumption of the exponential growth. The growth rate $r$ was estimated by incorporating the negative binomial likelihood as follows (*Ma, 2020*):

$$c_t \sim \text{NegBinom}(\text{mean} = i_0 e^{rt}, \text{overdisp.} = \phi), \quad (7)$$

where $t_{lo} \leq t \leq t_{up}$ and the parameters $\{i_0, r, \phi\}$ were supported by the following weakly-informative priors:

$$\log i_0, \log r \sim \text{Normal}(\text{mean} = 0, \text{SD} = 1),$$

$$\phi \sim \text{Gamma}(\text{shape} = 1, \text{scale} = 1). \tag{8}$$

Here, the negative binomial likelihood Eq. (7) was chosen over the Poisson likelihood in order to account for overdispersion in case counts reflecting the effect of possible processing errors.

$R_e$ was computed based on Wallinga and Lipsitch's formula (*Wallinga & Lipsitch, 2007*), which links $R_e$, $r$, and gamma-distributed generation time with the mean $\mu$ and SD $\sigma$:

$$R_e = \left(1 + r^\beta\right)^\alpha, \tag{9}$$

where $\alpha = (\mu/\sigma)^2$ and $\beta = \sigma^2/\mu$ are the shape and scale of the underlying gamma distribution. For this study, we used the serial interval as a proxy of the generation time and adopted the values $\mu = 10.1$ days and $\sigma = 6.1$ days, as estimated in the study by *Miura et al. (2023)*. We considered alternative estimates (*Guo et al., 2023*; *Guzzetta et al., 2022*; *Madewell et al., 2023*) for our sensitivity analysis.

### Extrapolation of case counts beyond 5 July 2023

Acknowledging the potential for variations in reported cases due to potential delays or modifications in reporting, we extrapolated the case counts from 5 July to 1 September 2023 under the assumption of sustained exponential growth. The case counts ($\bar{c}_t$), which encompass both previously reported and extrapolated cases based on symptom onset date followed the following equation:

$$\bar{c}_t \sim \text{NegBinom}(\text{mean} = i_0 e^{rt}, \text{overdisp.} = \phi), \quad t > t_{up},$$
$$\bar{c}_t = c_t, \quad t \leq t_{up}. \tag{10}$$

Here, "$\sim$" denotes sampling from the negative binomial distribution. The cumulative counts of cases for July and August were derived by aggregating $\bar{c}_t$ values across their respective date ranges.

For computing case counts based on the date of reporting ($n_t$), the counts ($\bar{c}_t$) were adjusted by considering the reporting delay. For every time $t$, the following sampling was conducted:

$$\{n_{t+s,t}; s > 0\} = \text{Multinomial}(\text{size} = \bar{c}_t, \text{probs} = \{f_t^*(s \mid \theta^*)\}), \tag{11}$$

where $f_t^*$ refers to Eq. (5). Thereafter:

$$n_t = \sum_{\tau \geq 0} n_{t, t-\tau}. \tag{12}$$

Finally, the aggregated count of the number of cases ($n_t$) for July and August was determined using summation over the respective date ranges.

### Global trends across the 2022 outbreaks

Mpox case count data for 2022 was collated from the WHO dashboard (*World Health Organization, 2023*). From this dataset, we selected 16 countries that confirmed over 700 symptomatic cases by the end of the year. Most of these nations provided daily case counts categorized by the date of symptom onset, the date of diagnosis, or the date of reporting. However, six counties in the WHO Region of Americas (AMRO)—Argentina, Brazil, Chile, Colombia, Mexico, and Peru—presented only aggregated case counts without any subtyping. For each country, we identified the exponential growth phase as a 45-day window that led up to the epidemic's peak, which was represented by the highest daily case count for 2022. In situations in which multiple peaks were recorded, we selected the earliest occurrence. In our sensitivity analysis, we considered alternating windows of 30 and 60 days.

For simplicity, we applied the reporting delay distribution from the Hong Kong data—a reasonable approach given its close alignment to a globally reported median of 6 days, an inter-quartile range spanning 4–9 days (*World Health Organization, 2023*), and the absence of more detailed global data. Using the methods described earlier, we then estimated the $R_e$ for each of the selected countries.

For the selected set of countries ($k = 1 \ldots K$), the pooled mean estimate $\hat{R}_e$ was calculated using a random-effects meta-analysis model (*Higgins & Thompson, 2002*). Each country's individual estimate of the mean $R_e^{(k)}$, was modeled to vary around the pooled mean $\hat{R}_e$ with two levels of variation accounting for an inter-specific, $\varepsilon_k$, and, second, intra-specific error, $\tau$:

$$
\begin{aligned}
R_e^{(k)} &\sim \text{Normal}\left(\text{mean} = \hat{R}_e^{(k)}, \text{SD} = \varepsilon_k\right), \\
\hat{R}_e^{(k)} &\sim \text{Normal}\left(\text{mean} = \hat{R}_e, \text{SD} = \tau\right).
\end{aligned}
\tag{13}
$$

Here, the parameter $\tau$ quantifies the between-country heterogeneity in $R_e$ estimates.

### Technical details

We adopted the Bayesian framework to fit each realization of the daily case counts classified by the date of symptom onset Eq. (6) to models Eqs. (7)–(9) using Markov chain Monte Carlo (MCMC) sampling techniques. The Bayesian estimation process was performed using Stan software (*Stan Development Team, 2023*). The inherent structure of the reconstructed Eq. (6) was challenging given its reliance on sampling back-projected counts from a multinomial distribution (Eq. 4). Because these counts, which were represented as integers, were not immediately compatible with the efficient use of Hamilton Monte Carlo sampling in Stan, we employed an approach similar to that previously described (*Weber et al., 2018*). For each realization of Eq. (4), we simulated the MCMC chain using 1,000 iterations for the tuning-in process and retained only a single posterior for further analysis. To construct the posterior distribution, we used 4,000 different chains that resulted in 4,000 posterior draws. The convergence of the MCMC simulation was inspected visually and was checked implicitly by fitting the model only for symptomatic cases. In doing so, the back-projection was not needed. One thousand

iterations that were used for tuning-in and four chains that were composed of 1,000 posterior draws resulted in an R-hat statistic below 1.01 (*Vehtari et al., 2021*). The code is made available on https://github.com/aakhmetz/Mpox-in-MainlandChina-2023.

## RESULTS

Using the Hong Kong data, we estimated a mean delay of 3.4 days (95% credible interval (CrI) [2.7–4.4]) from symptom onset to case diagnosis and a mean delay of 6.0 days (95% CrI [5.1–7.0]) from symptom onset to case reporting. Upon integrating these delays into the case back-projection (Fig. 1), the $R_e$ was estimated at 1.57 (95% CrI [1.38–1.78]) based on estimated serial interval (SI) distribution in *Miura et al. (2023)*, while the posterior mean ranged 1.31–1.76 using alternative SI estimates (*Guo et al., 2023*; *Guzzetta et al., 2022*; *Madewell et al., 2023*) (Table 1). Correspondingly, the doubling time was 15 days (95% CrI [11–21]). Projecting the mpox case counts past 5 July 2023 to cover the months of July and August, the mean count of cases that were categorized by the symptom onset date was 1,080 (95% CrI [650–1,720]) for July and 5,200 (95% CrI [1,820–12,060]) for August. Adjusting these numbers for the reporting delay, the mean numbers of reported cases were 800 (95% CrI [540–1,180]) and 3,850 (95% CrI [1,490–8,290]) for July and August, respectively.

The analysis of national outbreaks from selected countries in 2022 revealed a range in $R_e$ values. Portugal had the lowest mean $R_e$ value at 1.13 (95% CrI [1.06–1.23]), whereas Colombia had the highest value at 2.31 (95% CrI [2.04–2.60]), as illustrated in Fig. 2. The pooled mean $R_e$ was 1.53 (95% CrI [1.38–1.69]), with a between-country variance (heterogeneity) of $\tau^2 = 0.10$. However, upon evaluating the data quality provided to the WHO by individual countries, it became evident that the nations within the WHO AMRO—with the exceptions of the United States and Canada—provided only aggregated case counts without subtyping them by date of symptom onset, date of diagnosis, or date of reporting. Such reporting likely introduced higher variability and potential bias in the $R_e$ values. By excluding these countries from the analysis, the pooled mean $R_e$ dropped slightly to 1.49 (95% CrI [1.32–1.67]), and the between-country variance declined to $\tau^2 = 0.08$. Notably, the $R_e$, value for mainland China remained similar to the pooled average regardless of whether AMRO countries were included in the analysis.

The sensitivity analysis confirmed the robustness of our findings but highlighted some variability. When considering only cases with a known symptom onset date, the $R_e$ for mainland China remained consistent at 1.56 (95% CrI [1.37–1.79]), suggesting that most cases classified by date of reporting were predominantly linked to Taiwan. Varying the time window for the identified exponential growth phase—in which the lower bound $(t_{lo})$ varied between 16 and 26 May 2023, and the upper bound $(t_{up})$ varied between 30 June and 10 July 2023—the median posterior $R_e$ ranged from 1.51 to 1.57 (95% CrI [1.31–1.82]; Fig. S3). The posterior predictions for reported cases ranged from 650 to 840 (95% CrI [480–1,400]) for July and 2,150 to 3,540 (95% CrI [1,110–9,690]) for August (Figs. S4, S5). For the selected 2022 outbreaks, adjusting the length of the exponential growth phase to either 30 or 60 days did not critically alter the prior estimates (Fig. S6). Importantly, when we focused solely on cases with a known date of symptom onset, the estimated $R_e$ remained

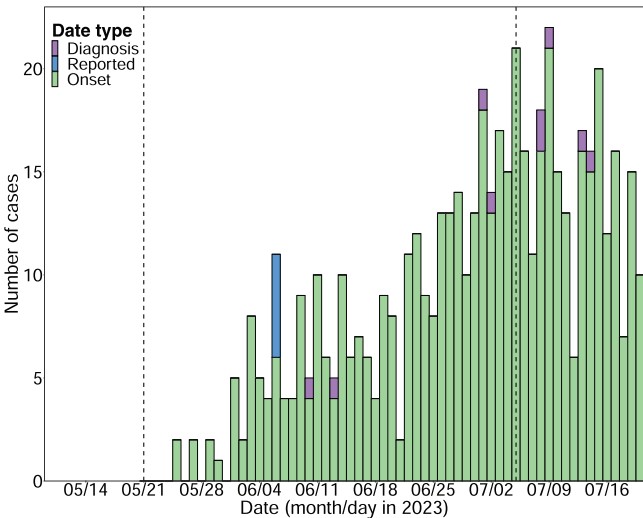

**Figure 1 The estimation of the case back-projection for mainland China, integrating reporting delays.** The exponential growth phase with dashed lines, representing the lower bound on 21 May 2023, and the upper bound on 5 July 2023.     

**Table 1 Posterior mean and 95% credible interval (CI) for the effective reproduction number, $R_e$, estimated for mainland China during the exponential growth phase and depending on the serial interval (SI) distribution.**

| Study | Mean SI, days | Standard deviation SI, days | Mean (95% CI) $R_e$ |
|---|---|---|---|
| *Miura et al. (2023)* | 10.1 | 6.1 | 1.57 [1.38–1.78] |
| *Guzzetta et al. (2022)* | 12.5 | 5.7 | 1.77 [1.50–2.07] |
| *Guo et al. (2023)* | 5.6 | 1.5 | 1.31 [1.21–1.42] |
| *Madewell et al. (2023)* | 8.5 | 5.0 | 1.47 [1.32–1.64] |

unchanged, indicating the minimal impact of case records with unknown symptom onset on the $R_e$ for the selected countries.

## DISCUSSION

In our analysis, we estimated that the $R_e$ of the mpox epidemic in mainland China was within the range of 1.3–1.8, depending on the chosen serial interval distribution (*Guo et al., 2023*; *Guzzetta et al., 2022*; *Madewell et al., 2023*; *Miura et al., 2023*). This range suggests a doubling time of approximately 2 weeks. Importantly, our estimate was based on the early phase of the epidemic, in which an evident exponential growth of cases was observed and control measures had likely not yet been implemented. Given the relatively low case counts of mpox and no deaths reported during the epidemic's early phase, significant behavioral changes in at-risk populations were likely not yet in effect, implying that mpox transmission occurred largely unhindered within affected communities during this early phase. Moreover, our projections for July-August, extrapolated into later months, might overestimate actual cases, as they do not account for the anticipated

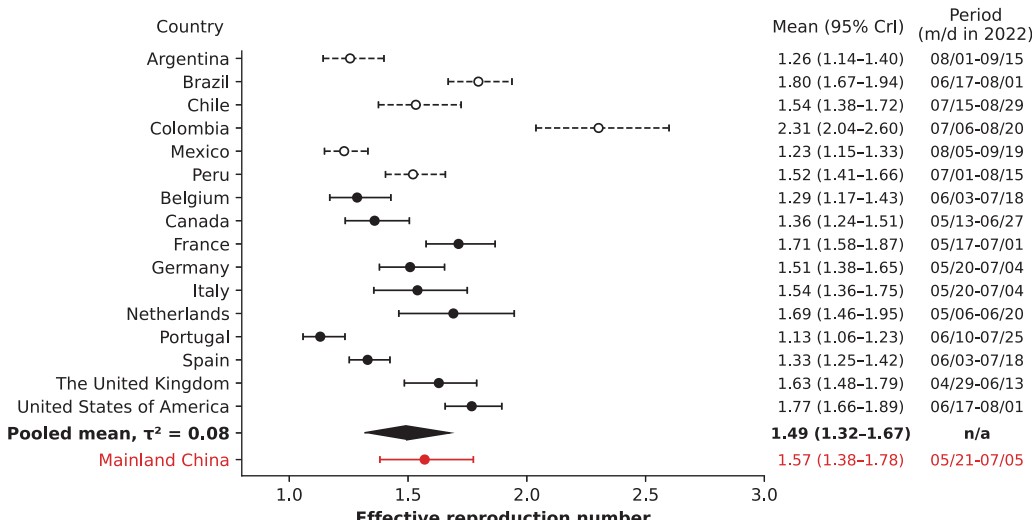

**Figure 2** Estimated effective reproduction number, $R_e$, for mainland China (red) comparing to selected national outbreaks of 2022 (black) and their pooled mean (black diamond). The $R_e$ values are shown in the first column on the right, while a time period of 45 days is indicated in the last column. Countries that have not reported case counts, subtyped by date of symptom onset, diagnosis date, and reporting dates, were excluded from calculation of the pooled mean and are shown in dashed black.

behavioral changes expected from rising public awareness and new information about the outbreak (*China Centers for Disease Control, 2023a*).

The $R_e$ value estimated in our study closely aligns with values derived from selected national outbreaks in 2022, suggesting the potentially universal character of the outbreaks. However, an accurate estimation of infections remains challenging given the likely underreporting of mpox cases. Several factors might contribute to this under-ascertainment. One reason may be the societal stigmatization associated with the disease that leads to individuals' reluctance to disclose their infection for fear of facing societal prejudice or discrimination. Another possibility is that infections can result from intimate encounters outside of a committed relationship, leading individuals to withhold their health status because of the fear of repercussions in their personal lives.

Our study has several limitations. First, we did not differentiate between indigenous and imported infections in our dataset, a factor that may have influenced the estimation of $R_e$. However, by focusing exclusively on the exponential phase in which case growth is most evident, the influence of imported cases on our estimates was likely minimal. Second, the actual method of imputing cases from Taiwan in the WHO dataset may have differed from ours or could have been modified by the WHO over time. For example, the observed pattern of data imputations by date of reporting that were aggregated every 7 days was no longer feasible at the end of July and August. Third, the reporting delay distribution was estimated using Hong-Kong line list data, which may differ from the data from mainland China given the differences in healthcare systems, case ascertainment, and case investigation protocols. Fourth, the results of comparisons of $R_e$ values across different countries must be interpreted with caution. The proactive control measures adopted by some nations *versus*

the more passive strategies employed by others can introduce variances. For instance, the outbreak in Portugal evolved at a slower pace given the government's proactive position. The case counts were almost steady over 2–3 months before a decline was observed. Although the $R_e$ values in our study and that of Du et al. (2022) were close to 1, a granular look at the case counts combined with a time-varying $R_e$ revealed a brief $R_e$ peak of 2.7 on 10 May 2022 that then gradually declined to 1 and dropped below this threshold by the end of June 2022 (Borges et al., 2023). Our mathematical framework successfully captured this average trend across Portugal and other countries, and a pooled mean $R_e$ of 1.66 summarized our overall findings. Finally, the WHO dataset retrieved on 15 August 2023 contained incomplete case counts given that some cases were yet to be reported. At the time of writing the results of this study, a snapshot for 12 September 2023 became available. The $R_e$ estimate only slightly changed to 1.63 (95% CrI [1.43–1.86]); however, the case counts for May and June were updated and revealed possible long reporting delays (Figs. S1A and S7).

Although mpox may be associated with a lower mortality rate compared with diseases such as smallpox, its morbidity is notably high, and mpox is associated with health complications such as pneumonia, confusion, and eye infections (Patel & Patel, 2023). Mpox disproportionally affects people living with human immunodeficiency virus (HIV) (Mitjà et al., 2023). The transmission of mpox throughout mainland China in the summer of 2023 was largely unconstrained given that regulatory guidelines were first published only at the end of July. This timeline put vulnerable populations at heightened risk of infections and subsequent complications. Nevertheless, the potential impact of efficient vaccination campaigns and information programs aimed at raising awareness of the disease cannot be overlooked; such interventions could significantly reduce case numbers and prevent a larger future outbreak. In contrast to mainland China, Taiwan managed to stabilize its mpox situation in 2023, maintaining an $R_e$ value of approximately one. Drawing from its previous experience—specifically, its management of the hepatitis A virus (HAV) outbreak among MSM in 2015–2017 (Lin et al., 2019)—the Taiwan government promptly initiated a vaccination campaign, targeted at high-risk groups that was bolstered by awareness campaign. The strategy not only ensured low vaccine hesitancy but also prevented an escalating outbreak. Mainland China could adopt a similar approach with vaccination campaigns focused on high-risk social groups coupled with the dissemination of accurate health messages. Avoiding stigmatization of the disease, an oversight during the early stages of the outbreak (Murphy, 2023; Yang, 2023), could further contribute to higher vaccination rates.

## CONCLUSIONS

In this study, the mean $R_e$ value estimated for the early phase of the mpox outbreak in mainland China was between 1.3 and 1.8, closely aligning with the pooled mean $R_e$ from selected outbreaks in 2022. This alignment suggests that mainland China could experience a similar trajectory of case growth as observed in other countries previously affected by mpox. These findings underscore the urgency for implementing immediate and effective

control measures, including targeted vaccination coupled with information campaigns to reduce stigmatization of vulnerable groups associated with the disease.

## ACKNOWLEDGEMENTS

We thank Anahid Pinchis from Edanz for editing a draft of this manuscript.

### Funding
This study was supported by the National Science and Technology Council, Taiwan (NSTC #111-2314-B-002-289). The funders had no role in study design, data collection and analysis, decision to publish, or preparation of the manuscript.

### Grant Disclosures
The following grant information was disclosed by the authors:
National Science and Technology Council, Taiwan: NSTC #111-2314-B-002-289.

### Competing Interests
The authors declare that they have no competing interests.

### Author Contributions

- Andrei R. Akhmetzhanov conceived and designed the experiments, performed the experiments, analyzed the data, prepared figures and/or tables, authored or reviewed drafts of the article, and approved the final draft.
- Pei-Hsuan Wu performed the experiments, analyzed the data, prepared figures and/or tables, authored or reviewed drafts of the article, and approved the final draft.

### Ethics
The following information was supplied relating to ethical approvals (*i.e.*, approving body and any reference numbers):

This study was based on publicly available data and did not require ethical approval.

### Data Availability
The dode and data are available at Zenodo: Akhmetzhanov, A. R., & Wu, P.-H. (2024). Data and code scripts for "Transmission potential of mpox in Mainland China, June-July 2023: estimating reproduction number during the initial phase of the epidemic" by Akhmetzhanov AR, Wu PH. In Transmission potential of mpox in Mainland China, June-July 2023: estimating reproduction number during the initial phase of the epidemic. Zenodo. https://doi.org/10.5281/zenodo.10468456.

### Supplemental Information
Supplemental information for this article can be found online at http://dx.doi.org/10.7717/peerj.16908#supplemental-information.

# PeerJ

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
