# Peer review of "Transmission potential of mpox in mainland China, June-July 2023: estimating reproduction number during the initial phase of the epidemic"

_PeerJ, doi:10.7717/peerj.16908_

## Round 0.1 · original submission · Minor Revisions

Congratulations on getting this far, if you can please address the minor revisions that the reviewers have recommended, it should be ready to be accepted. Please complete the minor revision within the next 21 days and resubmit. Best of luck!

Reviewer 1 ·

Basic reporting

The study by Akhmetzhanov et al. presents a thorough analysis of effective reproduction numbers during the initial phase of the mpox outbreak in China. The authors collected case data and relevant information from different sources and constructed more reliable epi curves to estimate effective reproduction numbers. Overall, the analysis that the authors conducted is well-described, and the manuscript provides timely insights for public health responses in China and other surrounding countries. There are several minor points that need to be clarified, and several arguments on the generalizability of their findings need to be reconsidered. Please find the attached review report.

Experimental design

Please see my comments in the review report.

Validity of the findings

Please see my comments in the review report.

Additional comments

Please see my comments in the review report.

Annotated reviews are not available for download in order to protect the identity of reviewers who chose to remain anonymous.

Reviewer 2 ·

Basic reporting

Please speficy what is meant by heterogeneity (tau^2) in the method section.

Experimental design

If the case count (c_t) refers to the number of onsets, the word "incidence" should not be used. Incidence refers to the true number of infections per day, not the number of symptom onsets. Additionally, when computing R0 as a function of the growth rate (r), you need to state that you assumed serial interval (time lag between onsets) as a substitute of generation time (time lag between infections).

Next, in formula 7, why do you use negative binomial, not Poisson, for likelihood estimations? Is it because "if the process error is not completely negligible, then choosing an overly dispersed distribution, such as the negative binomial distribution may be desirable" as mentioned in reference 24? If so, please mention the reason.

Validity of the findings

For reporting biases on May 2, 9, 16, etc, is it relevant to fit to a probability function? That is, the reporting biases in this case are not due merely to the reporting delay, which occurs at random, but they are caused by weekday biases or seasonal biasis, which change over time and may result in skewedness the delay distribution.

Additional comments

No comment to add

Reviewer 3 ·

Basic reporting

This is an excellent work. My comments are minor. It seems the authors used a specific generation time (GT) with mean 12 days and sd 5.7 days. I wonder whether other values should be considered.
For instance, Guo Z et al Journal of Medical Virology 95(01), 2022 estimated serial interval 5.6 days.

In discussion, "the Re=1.6-1.8, this range suggests a doubling time of approximately 2
weeks. " I presume this statement is also dependent on the assumption of GT.

Experimental design

The authors used well designed methodology.

Validity of the findings

The findings is acceptable.

---

## Round 0.2 · accepted · Accept

Thank you for making the changes and responding to reach revieweer point by point. I have assessed the revision myself and I am happy with the detailed explanation and current version. This manuscript is now ready for publication.